# Comparison of LEBA and RULA Based on Postural Load Criteria and Epidemiological Data on Musculoskeletal Disorders

**DOI:** 10.3390/ijerph19073967

**Published:** 2022-03-26

**Authors:** Dohyung Kee

**Affiliations:** Department of Industrial Engineering, Keimyung University, Daegu 42601, Korea; dhkee@kmu.ac.kr; Tel.: +82-53-580-5319

**Keywords:** work-related musculoskeletal disorders, observational method, LEBA, RULA

## Abstract

Various observational methods have been developed and applied in industrial settings with the aim of preventing musculoskeletal disorders (MSDs). This study aimed to compare the Rapid Upper Limb Assessment (RULA), a representative observational method, and the Loading on the Entire Body Assessment (LEBA), a newly developed tool for assessing postural loads and their association with MSDs. The two methods were compared in various categories, including general characteristics, risk levels, postural load criteria, association with MSDs, influencing factors, and inter- and intra-rater reliabilities based on relevant previous studies. The results showed that compared to the RULA, the LEBA was better at evaluating various factors affecting postural loads and assessing musculoskeletal loadings, was better correlated with various postural load criteria, could predict the association with MSDs more accurately, and had higher inter- and intra-rater reliabilities. Based on these comparisons, it seems that the LEBA may be better than the RULA for estimating postural stress and predicting the association with MSDs.

## 1. Introduction

Musculoskeletal disorders (MSDs) are a set of injuries and symptoms affecting the osteomuscular system and associated structures, such as the bones, muscles, joints, tendons, ligaments, nerves, and the circulatory system [1]. Work-related MSDs (WMSDs) are the leading cause of occupational disabilities in industrialized countries [2,3]. Assessment of exposure to WMSD risk factors is regarded as a critical step in protecting workers in industries from developing WMSDs [4].

Observational techniques that aim to evaluate risk factors are more widespread in industries [5] because they (1) do not interfere with job processes; (2) do not require the use of expensive equipment for measuring the angular deviation of a body segment from the neutral position; (3) are user-friendly, applicable, and repeatable in various conditions; and (4) have higher validity and lower subjectivity than that had by self-reports, such as worker diaries, interviews, and questionnaires [6,7,8].

While many observational methods have been developed and applied for assessing risk factors for WMSDs, the Rapid Upper Limb Assessment (RULA) [9] has been the most frequently applied method in industries [10,11]. The RULA has been cited approximately 3500 times in relevant literature, which exceeds the citations of other observational methods [12]. Some comparative studies [13,14,15,16,17,18,19] claimed that the RULA might be the best system among three representative observational techniques, including the Ovako Working Posture Analysis System (OWAS) [20], RULA, and Rapid Entire Body Assessment (REBA) [21] for evaluating postural loads and the association with MSDs.

Many existing observational methods were mostly developed based on rankings provided by ergonomists and occupational physiotherapists or on subjective opinions by experienced workers rather than objective and consistent experimental results [9,16,20,21]. Recently, Kee [22] developed a novel observational technique, the “Loading on the Entire Body Assessment” (LEBA), mainly based on the experimental results obtained from the studies performed by the author and other researchers. In the study, the LEBA was validated using postural load criteria, including discomfort, compressive force at L5/S1, maximum holding time and % capable at the shoulder and trunk, and epidemiological data on MSDs [22].

A study that compares various categories of observational methods would be helpful in selecting an observational method that could most accurately quantify postural stress and estimate the association with MSDs. Contrary to the RULA, which was frequently compared with various observational methods in previous studies [5,13,14,15,16,17,18,19,23,24], few studies have compared the LEBA with other observational methods because of its recent development. Kee [22] asserted in a study that the LEBA, compared to other representative observational methods, may be a more useful tool for precisely quantifying postural loads, as well as determining the corrective actions required and the association with MSDs. Therefore, a comparative study is needed to prove the superiority and validity of the LEBA. This study aimed to compare and evaluate the LEBA and RULA based on various postural load criteria and epidemiological data on MSDs. Among several observational methods, the RULA and LEBA were chosen for this study because (1) the RULA has been widely applied in industries and is known to be one of the best tools for estimating postural loads [13,14,15,16,17,18,19], and (2) the LEBA has advantages, such as a high correlation with various postural load criteria and a strong relationship between the LEBA risk levels and WMSDs [22].

### RULA and LEBA

The RULA was developed by McAtamney and Corlett [9]. It provides a quick assessment of the loading on the musculoskeletal system due to postures of the neck, trunk, and upper limbs, muscle function, and external loads exerted. Based on the grand score of its coding system, four action levels, which indicate the level of intervention required to reduce the risks of injury due to physical loading on the worker, were suggested [9,14]:Action level 1: posture is acceptable if it is not maintained or repeated for long periods;Action level 2: further investigation is needed and changes may be needed;Action level 3: investigation and changes are required soon;Action level 4: investigation and changes are required immediately.

The LEBA was proposed by Kee [22]. It was based on discomfort and epidemiological data from previous research, from which posture classification and scoring systems of representative observational methods were adopted and modified. The LEBA reflects the effects of posture, external load, motion repetition, static loading, and coupling. The LEBA has more detailed posture classifications compared to representative observational techniques, such as the OWAS, RULA, and REBA. The LEBA classifies leg postures into 13 categories and subcategories, and classifies motion repetitions into five categories. In addition, three equations were provided according to the hand position for rating the effects of the external load or exertion. The degrees of the assessed harmfulness are grouped into four action categories, according to the urgency for the required workplace interventions:Action category 1: normal postures that do not need any corrective actions;Action category 2: postures that require further investigation and corrective changes during a subsequent regular check, but immediate intervention is unnecessary;Action category 3: postures that require corrective actions, including redesigning workplaces or working methods, within a short time;Action category 4: postures that require immediate consideration and corrective actions.

## 2. Materials and Methods

The comparison perspectives or categories adopted in this study, including general characteristics, risk levels, postural load criteria, association with MSDs, influencing factors, and intra- and inter-reliabilities, were based on a review article by the author [15]. The comparison-relevant references, their results, and raw data for the RULA and LEBA were obtained from the previous review article [15] and studies conducted by the author [13,14,16,17,22].

### 2.1. General Characteristics

The general characteristics summarized for the comparisons were based on the studies by Takala et al. [5], McAtamney and Corlett [9], and Kee [22]. McAtamney and Corlett [9] and Kee [22] developed the RULA and LEBA in their studies, respectively.

### 2.2. Postural Load Criteria

Regarding the postural load criteria adopted in this study, data of perceived discomfort were obtained from the study performed by Kee [13]. The independent variables used in the study were three levels of hand distance (arm reach [AR] of 40%, 70%, and 100%), four levels of hand height (shoulder height [SH] of 30%, 65%, 100%, and 120%), and four levels of external load (0.0, 1.0, 3.0, and 5.0 kg). AR was defined as the maximum horizontal distance from the tip of the middle finger to the wall when participants stood upright with their back against the wall. SH was defined as the vertical distance from the floor to the acromion in the upright position [25]. The dependent variable was whole-body discomfort, which was measured by the Borg CR10 scale [26].

The data on maximum holding times (MHTs) from the study by Kee et al. [17], which measured the MHTs and discomfort in 72 experimental postures, were used. Discomfort was measured at 60 s after the experiment initiation. The experimental postures were determined by three levels of hand distance (AR of 40%, 70%, and 100%), four levels of hand height (SH of 40%, 70%, 100%, and 120%), two levels of trunk rotation (0° and 30°), and three levels of external load (0.0, 1.5, and 3.0 kg).

Data on compressive forces at L5/S1, % capable at the shoulder and trunk, and postural loads by the RULA for the 48 and 72 experimental postures were obtained from Kee [13] and Kee et al.’s [17] studies, respectively. The compressive forces at L5/S1 and % capable at the shoulder and trunk were calculated using the 3-Dimensional Static Strength Prediction Program (University of Michigan, MI, USA). The postural loads of the 48 and 72 experimental postures [13,17] for the LEBA were calculated in this study by the author. The postural load criteria and postural loads for the LEBA and RULA were used in the correlation analysis, distribution of risk levels, and calculation of agreement rates between the LEBA and the RULA.

### 2.3. MSD Cases

The association with MSDs was investigated using 209 real cases of MSDs and corresponding postures, which were adopted from the study performed by Kee [14]. The 209 cases were MSDs that occurred in the upper body, including the lower back. MSDs of the lower limbs were excluded from analysis because the RULA does not have enough postural classifications to appropriately assess postural loads of the lower limbs. In Korea, workers with MSDs, who are diagnosed by medical doctors, can apply to the Korea Workers’ Compensation & Welfare Service (COMWEL) for medical treatment for their industrial accident. The COMWEL determines whether the MSD of the applicants is work-related (i.e., WDSDs or MSDs attributed to factors other than work-related factors). The work-relatedness of the applicants’ MSDs is decided by two ergonomists or industrial medicine doctors recommended by the COMWEL, which was based on the guidelines for determining occupational diseases of MSDs. The MSDs were reported by workers in automotive, automotive parts manufacturing, and construction industries. Of the 209 MSD cases, 148 applications were approved as WMSDs by the COMWEL, while for 61 applications, their MSDs were attributed to factors other than work-related factors.

A posture that reflected the most awkward or stressful task among tasks performed by each applicant based on videos and photographs was selected and evaluated by the RULA and LEBA. The risk levels for the postures evaluated by the RULA and LEBA were identical with those reported by Kee [14,22]. The results of the logistic regression analysis regarding the postural loads evaluated by the RULA were the same as those reported by Kee [14]. The logistic regression analysis for the LEBA was newly performed in this study by the author. The LEBA score and action category and the RULA grand score and action level were used as independent variables, which were significant in the chi-square analyses. The dependent variable was whether an MSD was approved to be work-related or not (i.e., WMSDs vs rejected MSDs). Four separate logistic regression analyses were performed for each independent variable. To increase the reliability of the analyses, five LEBA action category 1 classes were recategorized as action level 2. Here, the LEBA score and the RULA grand score were assumed to be the interval scale, with the characteristics of continuous numbers.

### 2.4. Influencing Factors

Factors that influence the RULA grand score were obtained from the study performed by Joshi and Deshpande [27], who investigated the factors that could influence the evaluation of RULA using ordinal regression analysis, where the scores for the force or load and muscle use were set as 0. For comparison, the author conducted a regression analysis for the 209 MSD cases included in this study. The LEBA score was used as the dependent variable, and the scores for the body segments, including the wrist, elbow, shoulder, neck, trunk, and leg, as well as the external load, motion repetitions, static loading, and coupling, were used as the independent variables.

### 2.5. Intra- and Inter-Rater Reliabilities

The inter- and intra-rater reliabilities of the LEBA were based on the study by Kee [22], and those of the RULA were obtained from six previous studies identified through a literature review [9,28,29,30,31,32].

### 2.6. Statistical Analysis

Several statistical analyses were adopted for comparisons performed in this study. The Wilcoxon signed-rank test was used to test the significance of differences between the risk levels of the LEBA and RULA. The relationships between the LEBA and RULA and the postural load criteria were investigated through correlation analyses. The association with MSDs was examined using the chi-square test and logistic regression analysis. A regression analysis was conducted to identify factors influencing the LEBA scores. The significance level for this study was set at an α level of *p* < 0.05. All statistical analyses were performed using SAS 9.4 (SAS Inc., Cary, NC, USA) and Microsoft Excel (Microsoft Co., Redmond, WA, USA).

Most comparison data and analyzed results were obtained from previous studies [9,13,14,17,22,27,28,29,30,31,32], but some data and analyzed results were supplemented or calculated by the author for the comparisons performed in this study. For example, the postural loads by the LEBA for the experimental postures used in the studies by Kee [13] and Kee et al. [17] were newly obtained in this study, and the chi-square tests and logistic regression analyses of the LEBA scores and action categories for the 209 MSD cases were performed in this study. Some correlation analyses between the postural load criteria and the RULA grand score were also conducted in this study.

## 3. Results

### 3.1. General Characteristics

The general characteristics of the LEBA and RULA, which were based on previous studies [5,9,22], are summarized in Table 1. In assessing postural loads, the LEBA and RULA evaluate the same body parts, including the upper arm (or shoulder), lower arm (or elbow), wrist, neck, trunk, and leg, but the LEBA has more detailed posture classification categories. For example, while the RULA has four categories each for the upper arm and trunk, the LEBA has seven categories each. The LEBA has four postural categories, including standing, squatting, sitting, and kneeling, and a total of thirteen subcategories for leg postures, compared with the RULA that only has two categories based on whether the legs and feet are well-supported and are in an evenly balanced posture. The RULA’s posture classification scheme was developed based on the results from previous studies [9], while the LEBA classifies the body joint motions into categories based on more consistent and objective data on discomfort obtained through laboratory experiments. The RULA has four categories for force or external load, with scores assigned according to the size of the force or external load, while the LEBA provides three equations for the external load based on its size as well as the hand position, which is determined by the hand distance and height.

The RULA classifies the loads by motion repetition into only two categories, according to whether or not a posture is repeated more than four times per minute. The LEBA classifies the loadings into five categories based on the number of motion repetitions per minute that are assigned numerical discomfort scores, with the characteristics of the ratio scale. The LEBA and RULA similarly divide the loads by muscle use or static loading into two categories, but they each assign scores differently. The LEBA additionally assesses coupling effects. The LEBA and RULA adopt the same estimation method to evaluate only the left or right side that is under greater stress. If it is difficult to decide which side is more loaded, both sides are assessed. The LEBA and RULA do not specify an observation strategy, such as time sampling [5,9,22]. However, in general, the LEBA and RULA observe the most common, prolonged, or loaded posture. The two methods are equipped with four action levels/categories for classifying risk levels. While the four action levels of the RULA do not have a basis for classification, those of the LEBA are classified by the working posture classification based on the MHTs reported by Miedema et al. [25] and were modified after considering the results obtained by applying the LEBA to the 148 MSD cases approved as WMSDs by the COMWEL.

### 3.2. Risk Levels

The distribution of action categories/levels by the LEBA and RULA for postures used in the studies by Kee [13], Kee et al. [17], and Kee [22], is presented in Table 2. The LEBA showed higher postural loads for corresponding postures than the RULA in the three previous studies. However, the Wilcoxon signed-rank tests were statistically significant for only the postural loads reported in the studies by Kee [13] and Kee [22] (*p* < 0.01). The proportions of the postures with high postural loads requiring fast or immediate corrective actions (i.e., action categories and levels 3 or 4) were higher when evaluated by the LEBA (68.8%, 76.4%, and 77.7%) than by the RULA (62.5%, 58.3%, and 64.9%). In total, while approximately 75.7% of the assessments by the LEBA corresponded to action categories 3 and 4, approximately 62.7% of those by the RULA corresponded to action levels 3 and 4.

The agreement rates between the two methods were 54.2% in Kee [13], 59.7% in Kee et al. [17], and 59.5% in Kee [22]. The average agreement rate for the three studies was approximately 58.6%.

### 3.3. Postural Load Criteria

To evaluate the validity of the developed methods, the relationships between the LEBA and RULA grand scores and their representative postural load criteria were analyzed using correlation analyses [13,17,22]. The criteria included discomfort, MHTs, compressive force at L5/S1, and % capable at the shoulder and trunk. The correlation coefficients between the LEBA scores/RULA grand scores and the postural load criteria are presented in Table 3. Apart from the correlation coefficient between the compressive force at L5/S1 and the LEBA and RULA grand scores in Kee’s study [13], the correlation coefficients between the LEBA scores and other postural load criteria in Kee’s study [13], as well as all the postural load criteria in Kee et al.’s study [17], were higher than those observed between the RULA grand scores and the postural load criteria.

The Spearman correlation coefficients between the LEBA action category and the RULA action level were obtained based on the risk levels evaluated by the RULA and LEBA for the 48, 72, and 148 postures reported in the studies by Kee [13], Kee et al. [17], and Kee [22], respectively (Table 4). While the coefficients based on the studies by Kee [13] and Kee et al. [17] were high (>0.72), that based on the study by Kee [22] was relatively low.

### 3.4. Association with MSDs

The association with MSDs was investigated based on the 209 epidemiological data on the MSDs diagnosed by medical doctors [14,22] and was analyzed using the chi-square test and logistic regression analysis. These analyses revealed that the LEBA score (*p* < 0.01) and action category (*p* < 0.01), as well as the RULA grand score (*p* < 0.01) and action level (*p* < 0.05), were statistically significant determinants of WMSD. The logistic analyses indicated that a 1-point increase in the LEBA score and RULA grand score increased the odds ratio for the determination of an MSD as work-related by approximately 1.05 and 1.36 times, respectively (Table 5). The odds ratios for the LEBA action category and RULA action level were calculated with action category or level 2 as the reference, which was the minimum action category or level in the LEBA and RULA analyses. The LEBA action category 3 increased the probability that an MSD would be approved as work-related by more than two times (2.42) compared to that by the LEBA category level 2, while the LEBA action category 4 increased the probability by approximately seven times (7.00). The RULA action level 4 almost tripled the probability (2.56), compared to that by the RULA action level 2. The percentage concordant values of the logistic regression models for the LEBA score and action category were high (69.6% and 55.2%, respectively), while the values for the RULA grand score and action level were relatively low (52.4% and 44.8%, respectively).

### 3.5. Influences of Factors

Joshi and Deshpande [27] reported that the upper arm was the most influencing factor of the RULA score, followed by the trunk, neck, wrist, lower arm, and leg. The regression analysis performed in this study for the LEBA score demonstrated that the external load most significantly influenced the LEBA score, followed by the shoulder, trunk, leg, static loading, elbow, neck, motion repetition, coupling, and wrist.

### 3.6. Intra- and Inter-Rater Reliabilities

The intra-rater reliabilities of the agreement rates, ĸ values, and interclass correlation coefficient (ICC) for the LEBA were 80.0–100.0% (mean: 90.8%), 0.64–1.0 (mean: 0.83), and 0.98 (95% confidence interval: 0.98–0.99), respectively [22]. The inter-rater reliabilities of the agreement rates, ĸ values, and ICC for the LEBA were 75.0–95.0% (mean: 85.5%), 0.56–0.94 (mean: 0.75), and 0.97 (95% confidence interval: 0.95–0.98), respectively (Table 6).

A few studies reported on the inter- and intra-rater reliabilities for the RULA. McAtamney and Corlett [9] qualitatively concluded in their original study on the RULA that there was a high consistency among the scoring by 120 raters. The agreement rates for the inter-rater reliability reported by Breen et al. [28] and Widyanti [32] were 94.6% and 58.25%, respectively. Dockrell et al. [29] reported the intra- and inter-rater reliabilities for six raters based on the computed postures of twenty-four children. Based on their ratings, the ICCs of the intra-rater reliability ranged from 0.27 to 0.86 for the action levels and 0.47 to 0.84 for the grand scores, and those of the inter-rater reliability ranged from 0.54 to 0.72 for the action levels and from 0.50 to 0.77 for the grand scores. Laeser et al. [30] asserted that the inter-rater reliability using Kendall’s coefficient of concordance was statistically significant (Kendall’s W = 0.773). Oates et al. [31] reported in their study that the inter-rater reliability of the RULA was Ebel r = 0.73, based on the observations of children using computers. Widyanti [32] showed that the mean inter-rater reliability for the RULA was poor (ĸ value: 0.20).

## 4. Discussion

This study compared the efficiency of the RULA, one of the best observational methods [13,14,15,16,17], with that of the LEBA, a method that was recently developed by the author [22]. Of the two types of validation studies for observational methods, this study assesses concurrent validity, which evaluates the extent of correlation of a method (LEBA in this study) with more valid and established ones (RULA in this study), instead of predictive validity, which evaluates the extent to which scores accurately predict an item (in this study, the association of the risks estimated by the LEBA with MSDs) [5]. The comparisons were mainly based on various previous studies, including both types of validity studies (i.e., Kee [13] and Kee et al. [16,17]: concurrent validity; Kee [14]: predictive validity).

The results from the comparison suggest that the LEBA is a better observational method than the RULA in all categories compared in this study, including the general characteristics, risk levels, postural load criteria, association with MSDs, and intra- and inter-rater reliabilities. Although the author could not compare the LEBA with findings from a study that assessed the usability of the RULA, the LEBA exhibited high usability with “agree” or “strongly agree” (first and second highest level of five verbal anchors employed in the usability test) in the four perspectives evaluated in the usability test, such as “easy and time-effective”, “effective”, ”helpful to decide acceptance”, and “useful in establishing interventions” [22].

The agreement rates between the LEBA and RULA were low (<60.0%) in three studies [13,17,22]. The Spearman correlation coefficient between the LEBA action category and the RULA action level for the 148 WMSD cases from industries was also low (0.571) (Table 4). These results imply that the risk assessment results by the two methods do not agree or correlate well. However, it is inferred that the LEBA may evaluate postural loads more precisely than the RULA. This is because the correlation coefficients between the postural load criteria, which included discomfort, MHTs, compressive force at L5/S1, and % capable at the shoulder and trunk, and the LEBA score were generally higher than those between the postural load criteria and the RULA grand score (Table 3).

The logistic regression analysis revealed that the LEBA predicted the association with MSDs more accurately than the RULA, but the percentage concordant for the LEBA action category was <60% (the value for the RULA action level was 44.8%) (Table 5). This implies that when a practitioner decides whether or not an MSD is work-related, mainly based on the LEBA action category, almost half of those decisions may not be correct. This may be attributed to several reasons [22]. First, the RULA and LEBA do not appropriately assess work-related musculoskeletal loads, which are known to be the main risk factor for the development of MSDs [33,34,35]. Second, it is difficult to estimate the association with different MSDs in various body parts that were caused by several factors using a single ergonomic method. This suggests that it may be better to use several methods to rigorously evaluate musculoskeletal loadings and accurately predict WMSDs rather than using a single best technique. Thus, unlike most existing methods that are designed for the whole body and all MSDs, new techniques should be designed for specific body parts and/or MSDs [22]. This would enable health and safety practitioners in industries or ergonomists to assess musculoskeletal loadings more precisely, which would, in turn, increase the accuracy of predicting the association with MSDs.

While the odds ratios for the LEBA action category (2.42 and 7.00 for the LEBA action category 3 and 4, respectively) were excessively higher than those for the RULA action level (0.88 and 2.56 for the RULA action level 3 and 4, respectively), and the percentage concordant for the LEBA score (69.6%) was also excessively higher than that for the RULA grand score (52.4%), the odds ratio for the LEBA score (1.05 per 1 point) was lower than that for the RULA grand score (1.36 per 1 point) (Table 5). This may be attributed to the wide range of the LEBA scores (range: 2–63) compared to that of the RULA grand scores (range: 1–7) used for the logistic regression analysis performed in this study.

The influencing factors of the LEBA score were slightly different from those of the RULA grand score. If the factors such as the external load, motion repetition, static loading, and coupling, which were not considered in the study by Joshi and Deshpande [27], were excluded, the influence of the upper arm (or shoulder) on the LEBA and RULA grand scores was ranked first, and that of the trunk was ranked second. While the effects of the other factors were significant in the order of the neck, wrist, lower arm, and leg in the RULA grand score, the effects were significant in the order of the leg, elbow, neck, and wrist in the LEBA score. This difference may be attributed to the following facts: (1) the RULA was developed based on results from previous relevant studies, which can result in incorrect posture classifications without appropriately considering real postural loads, rather than the consistent experimental data used to develop the LEBA; (2) while the RULA has just two posture classification codes for the leg, the LEBA is equipped with four categories and thirteen subcategories for leg postures. The differences between the two methods may explain the inability of the RULA to properly reflect postural loads compared to the LEBA.

This study compared the RULA, which focuses on upper limb postures, and the LEBA, which was developed for evaluating whole-body postures. This might be justified based on the following studies. First, although the RULA has a significant limitation of comprising only two classifications for leg postures, many previous studies have assessed postural loads using the RULA, including even unstable lower limb postures, such as squatting and kneeling. Gómez-Galán et al. [12], after performing a bibliometric review of 226 RULA-relevant publications between 1993 and 2019, reported that the RULA was applied to manufacturing (74 studies), human health and social work activities (38), agricultural activities, forestry and fishing (18), construction (4), and mining and quarrying (2). Kee and Karwowski [16] also applied the RULA to iron and steel industries, and general hospitals. The above are representative industries where various unstable lower limb postures occur frequently. In principle, the RULA, with two classifications for leg postures, including balanced or unbalanced, could not be applied to the above industries. However, the aforementioned showed that the RULA has been applied to whole-body postures irrespective of leg postures. Second, while the RULA was developed for assessing upper limb postures [9], the OWAS and the REBA were developed for evaluating whole-body postures [20,21]. Kee [15] showed, based on a literature survey, that 44 journal papers dealt with the assessments of postural loads using 2 or more of the OWAS, RULA, and REBA, and that of the 44 studies, 39 adopted the RULA. Nine of the ten studies dealing with the OWAS and RULA applications revealed that the postural loads shown by the RULA were higher than those shown by the OWAS. Of the 36 studies that adopted the RULA and REBA as ergonomic risk assessment tools, 30 demonstrated that the RULA showed higher postural loads for the selected postures than the REBA. Third, several studies compared various observational techniques, including the three methods of the OWAS, RULA, and REBA, based on scales for posture classification, main functions, correspondence with valid reference, association with WMSDs, repeatability between observers, potential users, ergonomic experts’ evaluation, exposure factors assessed, postural loads, discomfort, maximum holding time, ergonomic experts’ evaluation results, etc. [5,7,11,13,14,15,16,17,18,19,23,24,32]. The LEBA was developed for estimating whole-body postures like the OWAS and REBA. These imply that the LEBA can be compared to the RULA, a representative observational method.

The results presented in this study should be interpreted with caution, because there are so many RULA-based studies or cases [12], but very few studies or cases are LEBA-based.

The comparisons in this study were based on general characteristics, risk levels, postural load criteria, association with MSDs, and intra- and inter-rater reliabilities. The data were obtained from laboratory experiments, as well as automotive manufacturing, automotive parts manufacturing, and construction industries. Further comparative studies using postures and MSD cases from various industries may be required to obtain more reliable comparison results.

## 5. Conclusions

The comparisons revealed the following advantages of the LEBA, compared to the RULA: (1) the LEBA was developed based on objective experimental data, rather than subjective data such as workers’ experience and experts’ judgements [9,22], and it had more detailed classifications, especially for the external load and lower limbs [9,22] (Table 1); (2) the LEBA showed higher risk levels in postural load estimations than the RULA (Table 2); (3) the LEBA scores were better correlated with the postural load criteria than those by the RULA (Table 3); (4) the LEBA predicted the association with MSDs more accurately than the RULA (Table 5); and (5) the intra- and inter-rater reliabilities were excessively higher for the LEBA than for the RUBA (Table 6). Based on these, it can be concluded that until more reliable and valid observational techniques are developed, the LEBA may be better than the RULA for assessing postural loads and the association with MSDs.

## Figures and Tables

**Table 1 ijerph-19-03967-t001:** General characteristics of the RULA and LEBA.

	Assessment Factors					Observation Strategy	Body Side Assessed	Risk Category
	Posture	Force/External Load	Motion Repetition	Static Action	Coupling			
RULA	Upper arms, lower arms, wrist, neck, trunk, leg	Four categories	Two categories	O	X	No detailed rules	Right or left side	Four action levels
LEBA	Shoulder, elbow, wrist, neck, trunk, leg	Three equations by three zones according to hand position	Four categories	O	O	No detailed rules	Right or left side	Four action categories

O: included; X: not included; LEBA: Loading on the Entire Body Assessment; RULA: Rapid Upper Limb Assessment.

**Table 2 ijerph-19-03967-t002:** Risk levels of the LEBA and RULA in each study.

Data Source	Method	Action Category/Level				
		1	2	3	4	Total
Kee [13]	LEBA	2 (4.2) *	13 (27.1)	14 (29.2)	19 (39.5)	48 (100.0)
	RULA	0 (0.0)	18 (37.5)	22 (45.8)	8 (16.7)	
Kee et al. [17]	LEBA	4 (5.6)	13 (18.0)	35 (48.6)	20 (27.8)	72 (100.0)
	RULA	4 (5.6)	26 (36.1)	16 (22.2)	26 (36.1)	
Kee [22]	LEBA	0 (0.0)	33 (22.3)	50 (33.8)	65 (43.9)	148 (100.0)
	RULA	0 (0.0)	52 (35.1)	37 (25.0)	59 (39.9)	
Total	LEBA	6 (2.2)	59 (22.0)	99 (36.9)	104 (38.8)	268 (100.0)
	RULA	4 (1.5)	96 (35.8)	75 (28.0)	93 (34.7)	

*: values within parentheses represent the percentage relative to the total number of assessments; LEBA: Loading on the Entire Body Assessment; RULA: Rapid Upper Limb Assessment.

**Table 3 ijerph-19-03967-t003:** Correlation coefficients between the LEBA and RULA grand scores and the postural load criteria.

Data Source	Postural Load Criterion		LEBA Score	RULA Grand Score
Kee [13]	Discomfort		0.864 *	0.554 *
	Compressive force		0.684 *	0.710 *
	% capable at	Shoulder	−0.637 *	−0.242
		Trunk	−0.762 *	−0.591 *
Kee et al. [17]	Discomfort		0.704 *	0.599 *
	MHT		−0.680 *	−0.649 *
	Compressive force		0.917 *	0.734 *
	% capable at	Shoulder	−0.608 *	−0.220 *
		Trunk	−0.724 *	−0.535 *

*: significant α = 0.01; LEBA: Loading on the Entire Body Assessment; RULA: Rapid Upper Limb Assessment.

**Table 4 ijerph-19-03967-t004:** Spearman correlation coefficients between the risk levels evaluated by the LEBA and RULA.

Kee [13]	Kee et al. [17]	Kee [22]
0.752 *	0.724 *	0.571 *

*: significant α = 0.01; LEBA: Loading on the Entire Body Assessment; RULA: Rapid Upper Limb Assessment.

**Table 5 ijerph-19-03967-t005:** Results of logistic regression analysis.

Independent Variable	N	OR	95% CI	% Concordant
LEBA score				
Continuous (per 1 point)	209	1.05	1.02–1.08	69.6
LEBA action category				
2	65	1		55.2
3	70	2.42	1.19–4.94	
4	74	7.00	2.99–16.38	
RULA grand score				
Continuous (per 1 point)	209	1.36	1.10–1.68	52.4
RULA action level				
2	76	1		44.8
3	62	0.88	0.44–1.78	
4	71	2.56	1.17–5.58	

OR: odds ratio; CI: confidence interval; LEBA: Loading on the Entire Body Assessment; RULA: Rapid Upper Limb Assessment.

**Table 6 ijerph-19-03967-t006:** Intra- and inter-rater reliabilities by studies.

Methods	Study	Applied Fields	No. of Raters	Intra-Rater Reliability	Inter-Rater Reliability
LEBA	Kee [22]	Automotive manufacturing and manufacturing of its parts, Construction	12	- % agreement: 80.0–100.0% (mean: 90.8%) - ĸ value: 0.64–1.0 (mean: 0.83) - ICC *: 0.98	- % agreement: 75.0–95.0% (mean: 85.5%) - ĸ value: 0.56–0.94 (mean: 0.75) - ICC: 0.97
RULA	McAtamney and Corlett [9]	Keyboard operations, packing, sewing, and brick sorting tasks	120	-	High consistency
	Breen et al. [28]	Computer workstation	3	-	94.6%
	Dockrell et al. [29]	Computer work environment	6	- 0.27–0.86 for the action levels - 0.47–0.84 for the grand scores	- 0.54–0.72 for the action levels - 0.50–0.77 for the grand scores
	Laeser et al. [30]	Computer workstation	-	-	Kendall’s W = 0.773
	Oates et al. [31]	Computer work environment	1	-	Ebel r = 0.73
	Widyanti [32]	Tofu, military equipment manufacturing, automotive maintenance and service, crackers, and milk processing	50		- % agreement: 58.25% - #x138; value: 0.20

*: interclass correlation coefficients; LEBA: Loading on the Entire Body Assessment; RULA: Rapid Upper Limb Assessment.

## Data Availability

The datasets used and/or analyzed during the current study are available from the corresponding author on reasonable request.

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
