# Peer review of "Comparison of LEBA and RULA Based on Postural Load Criteria and Epidemiological Data on Musculoskeletal Disorders"

_ijerph, 2022, doi:10.3390/ijerph19073967_

Round 1
Reviewer 1 Report
The article has improved significantly. Authors are recommended to reply explicitly to the reviewers in a separate document when uploading it again for next time.
Best Regards
Author Response
Dear Reviewer #1:
The author appreciates your very favorable review of my paper.
Reviewer 2 Report
I agree with all the changes made by the authors.
I take this opportunity to congratulate you for your work and dedication in improving the final article.
Author Response
Dear Reviewer #2:
The author appreciates your very favorable review of my paper.
Reviewer 3 Report
Dear authors,
the topic of the article is interesting but there are some aspects that I think need to be clarified.
In my opinion the main critical aspect both from a methodological point of view and in the discussion is the evaluation of epidemiological data on musculoskeletal disorders. Firstly, the criteria for the attribution of musculoskeletal disorders should be clarified, and secondly, it should be specified which musculoskeletal disorders are considered (disorders of the spine, shoulder, elbow, wrist, hip, knee, foot), also in view of the different roles of the biomechanical overload factors involved.
Round 2
Reviewer 3 Report
Dear author,
I greatly appreciate your work. The manuscript has been improved and I have suggested to the editor that it be accepted in its present form.